# “They Were Willing to Work with Me and Not Pressure Me”: A Qualitative Investigation into the Features of Value of a Smoking Cessation in Pregnancy Program for Aboriginal and Torres Strait Islander Women

**DOI:** 10.3390/ijerph18010049

**Published:** 2020-12-23

**Authors:** Vivian Lyall, Jillian Guy, Sonya Egert, Leigh-Anne Pokino, Lynne Rogers, Deborah Askew

**Affiliations:** 1School of Clinical Medicine, Primary Care Clinical Unit, Level 8 Health Sciences Building, Royal Brisbane & Women’s Hospital, The University of Queensland, Herston, QLD 4029, Australia; v.lyall@uq.edu.au; 2Southern Queensland Centre of Excellence in Aboriginal and Torres Strait Islander Primary Health Care, Queensland Health, P.O. Box 52, Inala, QLD 4077, Australia; jillianguy80@gmail.com (J.G.); Sonya.Egert@health.qld.gov.au (S.E.); Leigh-Anne.Pokino@health.qld.gov.au (L.-A.P.); Lynne.Rogers2@health.qld.gov.au (L.R.)

**Keywords:** Aboriginal and Torres Strait Islander women, pregnancy, smoking cessation, women-centered, social determinants, strength-based, culturally centered

## Abstract

With tobacco commonly used for stress relief, smoking cessation during pregnancy can present challenges for women facing stressful circumstances. This can be pronounced for Aboriginal and Torres Strait Islander women who experience disproportionately high smoking rates during pregnancy and also have a greater intersection of stressors from social disadvantage, institutional racism and trauma. To contribute understandings into how women can be best supported at this time, this study identified the features of value of an Aboriginal and Torres Strait Islander pregnancy smoking cessation program that addressed the contexts of women’s lives in culturally affirming and strength-based ways. A narrative methodology using a yarning approach was used to interview 7 pregnant women, 6 significant others, 3 case managers, and 4 healthcare professionals. Data were analyzed using thematic analysis, guided by an Indigenist research practice of deep and reflexive researcher listening. Features of value included: relationship-based care, holistic wraparound care, flexibility, individualized care, and culturally orientated care. Combined, they enabled highly relevant and responsive women-centered, trauma-informed, and harm-reducing smoking cessation support that was well received by participants, who achieved promising smoking changes, including cessation. This approach strongly departs from standard practices and provides a blueprint for meaningful support for pregnant women experiencing vulnerabilities.

## 1. Introduction

For women who smoke tobacco, pregnancy can present an immediate motivation to pursue smoking cessation. However, seeking support to address tobacco dependency at this time can be complex due to common experiences of external judgement and internal shame for potential harms imposed on their baby [1,2]. For Aboriginal and Torres Strait Islander women, intersecting social and structural barriers to their health and wellbeing during pregnancy, including systemic racism, impose additional challenges that can render the goal of cessation seemingly unrealistic or of lower priority, despite a desire to quit and knowledge of the adverse effects of tobacco smoking on their baby [1,3,4].

Tobacco smoking during pregnancy is one of the most important and modifiable risk factors leading to the unacceptable disparity in Indigenous maternal and infant health outcomes internationally [5]. In Australia, despite recent trends showing reductions in tobacco smoking initiation rates among Aboriginal and Torres Strait Islander youth [6], smoking levels remain high, particularly in pregnancy, whereby an estimated 44% of Aboriginal and Torres Strait Islander women smoked during pregnancy in 2018: a notable decrease from 52% in 2009, but still significantly higher than 9.6% observed among their non-Indigenous Australian peers [7]. 

Australia is not alone in this regard, with high rates of smoking during pregnancy among Indigenous women reported in many colonized countries, including Canada, United States of America, and New Zealand [8]. These high rates of smoking are interconnected with the complex stressors extending from historical, social, cultural, and economic determinants [9,10,11]. Smoking is linked to experiences of intergenerational and lived trauma from on-going colonization, the loss of land, and disruptions to Indigenous cultures, languages and families through aggressive assimilation policies, including the Stolen Generations in Australia and Residential Schools in Canada [12,13]. In Australia, the legacy of tobacco used as payment or issued as part of rations for those living on mission stations entrenched its use among Aboriginal and Torres Strait Islander peoples, and contributed to the current high rates of tobacco use [11,14,15]. Intersecting these factors, contemporary experiences of racism and socioeconomic inequities impacting all facets of life culminate in serious stressors for Aboriginal and Torres Strait Islander people, particularly for women who are pregnant [16]. In response, cigarettes are a commonly used mechanism for coping [4,11]. 

While social and structural barriers to Indigenous women’s reduction or cessation of tobacco use during pregnancy are understood globally [4,8,12,15,17], few interventions have sought to address these in the Australian context [18]. Rather, smoking cessation initiatives have largely favored individualistic behavioral approaches [18,19]. Failing to respond to the context of Aboriginal and Torres Strait Islander women’s tobacco use limits program capacity to address barriers to smoking cessation, while risking further stigmatization of those who do smoke during pregnancy and compounding a deficit view of Indigenous peoples [1,2].

A strengths-based approach provides an alternate paradigm for considering and addressing smoking cessation during pregnancy for Indigenous women [20,21,22]. Here, deficit views of Indigenous people’s health and wellbeing that are all too common in mainstream health promotion [23] are replaced with recognition and place for the inherent strength of Indigeneity in providing a solid foundation from which smoking cessation programs can be built. From this foundation, programs that recognize and support self-agency, foster supportive relationships at the family and community level, and facilitate non-judgmental access to the available resources and organizations to ameliorate the immediate impact of the social and structural determinants of health are likely to be effective, particularly when combined with smoking cessation support tailored to each individual’s needs [4,8,20,24]. Underpinned by this understanding, a holistic multifaceted smoking cessation in pregnancy program, the Empowering Strong Families (ESF) 4077 program, was developed and implemented in an urban Aboriginal and Torres Strait Islander community in Brisbane, Australia [25]. 

Details of the ESF program have been reported elsewhere [24], and therefore are summarized here. The program combined case management support, individualized and tailored smoking cessation support including nicotine replacement therapy (NRT), and an arts program to support smoking cessation for women pregnant with an Aboriginal and Torres Strait Islander baby (either the mother or father self-identified as Indigenous). In recognition of the social nature of smoking in Aboriginal and Torres Strait Islander communities, pregnant women were encouraged to have a significant other complete the program with them. Thirty-one pregnant women and 16 significant others participated in ESF. The acceptability and impact of the ESF intervention were determined through a mixed methods, exploratory study reported in detail elsewhere [25]. Over one-third of pregnant women (36% (4/11)) had quit at the three-month assessment, with two remaining smoke free one month postpartum. Nearly half the participants reported a quit attempt during the program, nearly all reported a reduction in the number of cigarettes smoked, and participants and health professionals all reported high levels of satisfaction with the intervention [25].

While the positive outcomes of ESF demonstrate its promising capacity for addressing Aboriginal and Torres Strait Islander women’s tobacco use during pregnancy [25], understanding the core features of the program that contributed to its success and enabled the program to be positively received by ESF participants and health professionals, would enable replication of the program in other settings. Therefore, this exploratory research aimed to identify the features of value (FOV) of the ESF program, as experienced by ESF participants, case managers and COE healthcare professionals, thereby providing critical insights for the development of similar projects in other locations. 

## 2. Materials and Methods 

### 2.1. Study Setting and Design 

ESF was implemented at the Southern Queensland Centre of Excellence in Aboriginal and Torres Strait Islander Primary Health Care (COE), a Queensland Government funded health service located in Inala (a southwestern suburb of Brisbane) that provides integrated primary and secondary health care predominantly to Aboriginal and Torres Strait Islander people [26]. Recruitment to the ESF program occurred from late November 2016 until mid-December 2017. Delivery of the ESF intervention continued until June 2018 when the non-recurrent funding ceased.

To facilitate our qualitative investigation into the FOV of the ESF program we used narrative inquiry as it enabled us to center the Aboriginal and Torres Strait Islander research methods of storytelling and yarning [27]. These methods are conversational approaches to qualitative data collection that center Indigenous ways of communicating and support in-depth discussion with participants and healthcare professionals in a culturally respectful manner [28]. Prioritizing researcher–participant relationships that honor participants’ agency and safety was foundational to the yarning process [29].

### 2.2. Interviewees and Data Collection

Using a convenience sampling strategy [30], seven pregnant women, six significant others, four COE health professionals and all three case managers were approached and agreed to participate in the interviews. Interviews were guided by the yarning process, which enabled participants to respond to the research topic on their own terms through informal conversation that moved between social yarning and research yarning [29]. In practice, this resulted in interviews commencing with social yarning, to prioritize participant-researcher relatedness and comfortability. Following this, the interviewer asked research related questions if conversations needed prompting or guidance to stay close to the research topic. To respect the privacy of pregnant women and their significant others, they were invited to participate by their case manager through a confidential conversation and were informed of their right to decline participation without explanation. Importantly, the researcher who conducted these interviews was already known to the pregnant women and their significant others through their visits to the COE. Given the personal nature of the research topic, pre-established relatedness was important for creating safety for participants to share their story. ESF case managers, two of whom were Indigenous, were interviewed at least once during the project. The interviewed COE health professionals all had patients in the ESF program, and included two general practitioners (GPs), the tobacco treatment specialist and a psychologist. Interviews ranged from 20 to 60 min, were conducted from April to November 2017, and with permission, were audio-recorded and transcribed verbatim. Transcripts were de-identified prior to analysis. While PW and significant others’ interviews did not reach data saturation due to the time constraints of the study, recurrent themes were clear across all interviews and therefore this did not limit our ability to identify and explore the FOV of the program.

### 2.3. Data Analysis

The data were analyzed in an iterative process using thematic analysis [31]. To guide this process we drew on the research practice described in *Dadirri—*an Indigenist research approach that calls for researchers’ deep listening for what is being communicated, along with what is not shared [28]. A commitment to our critical reflexivity for how we listened to and analyzed participants’ stories was pivotal to this process [28], as was mindfulness of the local, national, and historical contexts within which participants’ stories were being shared. Two researchers read and re-read the transcripts initially, and using NVivo 12, identified preliminary themes that illuminated the program’s FOV. These findings were then explored at an ESF team workshop, attended by the case managers, their line manager, the program coordinator, and three researchers. The team members reviewed the preliminary analysis and refined the FOV until consensus was reached. Throughout this process, team members drew on their experiences and understandings of ESF, which critically, were informed by the local Indigenous community knowledge and world views held by the Indigenous team members.

### 2.4. Indigenous Community Governance of the Research and Ethics

Throughout all stages, ESF was guided by and answerable to the Inala Indigenous community. Key Indigenous community and COE health professional staff, including two Indigenous members of the program team, along with ESF’s Indigenous Women’s Advisory Committee provided important guidance and feedback in the project’s development, implementation and evaluation. The Inala Community Jury for Aboriginal and Torres Strait Islander Health Research (the COE community research advisory group, consisting of members of the Inala Aboriginal and Torres Strait Islander community who endorse all research conducted through the service) provided community approval [32] and ethical clearance was obtained from the Metro South Human Research Ethics Committee (HREC/16/QPAH/597). Pregnant women and significant others gave written informed consent prior to their involvement in ESF, and additional consent was obtained from them and the health professionals prior to the interviews. 

## 3. Results

The features of value of ESF’s model of care were that it was relationship-based, holistic, flexible, individualized, and culturally orientated (Figure 1). Although the FOV are presented here as stand-alone concepts, in reality, they were highly interconnected, and each was equally important to participants. 

### 3.1. Relationship-Based Care: Safety to Engage, Safety to Disclose

Fostering trusted relationships between participants and their case managers was a fundamental FOV and supported participants’ on-going engagement in ESF. The case managers facilitated relationship building through non-judgmental, empathetic, and attuned deep listening to allow safety for participants to share personal and challenging aspects of their lives. Case managers also approached assessments without pressure, using informal yarning and allowing as much time as needed to gather information. This approach was highly valued by participants,

“I can trust her and she said to me I could tell her anything, which I did …”(pregnant woman)

“… instead of bagging me you listened to me and I just tell [my case manager] my problems and [my case manager would] help me out, which is good … that’s all I wanted …”(significant other)

Trusted relationships were facilitated by the continuity of care provided by case managers to pregnant women and significant others, and enabled participants to disclose personal issues that were previously unreported to other COE staff, despite most participants being prior COE clients. For example, participants disclosed current or past use of marijuana, alcohol, or methamphetamines, complex circumstances including domestic violence or homelessness, along with self-reports of depression or anxiety. These disclosures enabled case managers to understand participants’ unique circumstances and how to best direct support. Critically, case managers understandings enabled a more integrated approach to ESF participants’ healthcare within the COE, with case managers working collaboratively with other healthcare professionals to ensure a shared understanding of each participants’ unique needs. Integrating care ultimately increased the capacity of other healthcare professionals to improve the relevancy and quality of care they provided to ESF participants. However, the complexity and multiplicity of challenges experienced by ESF participants, however, had not been fully anticipated within the program design. Thus, case managers required psychoeducation and support to appropriately respond, as the COE psychologist reflected,

“… there’s actually been some pretty serious and complicated mental health presentations that we’ve had to [address] … [we have had to do] very quick, condensed, psychoeducation about things like borderline personality disorder, suicidal behaviors, how to manage those things, and appropriate referrals and all that kind of stuff because it’s been a very big, big adjustment for the staff to have to learn those things …”(Psychologist)

Finally, the dependability of case managers was a critical element in their relationship development, and participants cherished knowing that case managers would always follow through on promises of support. As one pregnant woman commented, “she was 100% there all the time, you know, helping me with getting anything …”. Experiencing trust and dependability was important for ESF participants, many of whom had few such previous experiences with people in their lives, including other service providers. As one significant other reflected,

“The people [case managers] are so nice and living in an area with lots of racism and say, not so much helpful people around, and just knowing there’s so much support, and there’s always someone there to give that little bit more guidance they need, and yes. I needed that at the time.”(significant other)

### 3.2. Addressing Barriers to Smoking Cessation through Holistic and Wraparound Care 

Smoking was commonly used by participants to mitigate stress associated with unmet basic needs such as shelter, safety and lack of necessities such as food, transport and clothing. As such smoking cessation was not a high priority for many participants. By considering participants within their social contexts, case managers provided wraparound care to co-create solutions with participants to address unmet needs. Ensuring their basic needs were met created the platform for participants to contemplate changing their smoking behaviors. As one case manager reflected,

“… [participants often say] I’ve got mental health, drug and alcohol [issues], why am I going to quit smoking when that’s how I survive … [so] it’s really, really difficult to get to the core of the smoking … you’ve got to cut through a lot of layers … I don’t think you can address any issues like smoking without addressing the social and emotional wellbeing … but if you can get them a house, get them a stable environment, get them food, get them [welfare] payments, you’re halfway there to helping the smoking …”(case manager)

Implementing a holistic approach required that case managers navigate across a great diversity of service areas. Many ESF participants were unaware of, and therefore not receiving the full complement of available health and social support they were entitled to and needed. Due to unmet basic needs combined with complex challenges, many also had reduced psychological capacity to navigate the complex bureaucratic processes to access available supports. With the physical and psychological support of the case managers, ESF participants were enabled access to supports and services, including: GP and specialist appointments for personal, antenatal, mental health and substance addiction support; government and non-government services for income, employment, housing and food assistance; legal services and assistance navigating the criminal justice system; and social and cultural supports through local Aboriginal and Torres Strait Islander organizations and childcare. The case managers frequently accompanied participants to appointments to provide advocacy and continuity, ensure psychological and cultural safety, and provide transportation if needed. The impact of this wraparound support was profound for many participants who achieved greater security and stability than they had previously known, as one pregnant woman shared, 

“… so I got this house basically because of her [case manager] … Centrelink she’s helped me with a lot. Tomorrow we’re going to do furniture for my house and she’s helped me a lot with my doctors and just the—because I’ve had a bit of a hard pregnancy … just with pain and stuff like that so she’s helped me a lot … yeah she’s taken me to get food vouchers and stuff like that …”(pregnant woman)

### 3.3. A Flexible Approach for Providing Highly Responsive Support

The complexity of challenges experienced by ESF participants resulted in everyday lives marked by a high degree of unpredictability due to frequent crises requiring immediate attention. In response, flexibility was essential and informed how case managers provided wraparound care and smoking cessation support. The COE psychologist commented, 

“… I like the fact that [the case managers] can be a bit flexible around what sort of issues they help the girls with because even though it’s primarily being young, pregnant, and smoking, there have been lots of other issues that have been picked up and they’ve actually been able to get some really good outcomes with housing and child safety and even Centrelink and making sure that the girls are linked in properly for follow up with the hospitals and all of those extra things that, normally, we wouldn’t have the time or the capacity to do.”(Psychologist)

A flexible approach was enacted through case managers reaching out to participants in their homes or elsewhere in the community, rather than restricting care to the health service. Enabling this, two health service vehicles were prioritized for ESF and fitted with baby seats, allowing case managers to readily visit ESF participants, and transport them to appointments. Furthermore, contact with ESF participants was not restricted by number or by time per contact, nor was support constrained by a structured, manualized program. Instead, case managers were highly responsive to ESF participants, assisting in meeting their immediate needs whilst keeping smoking cessation on the agenda, as one case manager highlighted,

“We’re not sitting at the desk [where] they have to come to us … we have that flexibility of going out into the community and being able to link in and engage with them [which] has been really helpful …”

A flexible approach to smoking cessation support was informed by an understanding that women do the best for their unborn child within their circumstances and capacity. Here, three elements of ESFs flexible support were particularly noteworthy. First, conversations about smoking reduction and cessation were informal and guided by what participants were ready to discuss. This approach enabled the focus of smoking reduction or cessation to remain on the table without judgement or pressure, as one case manager reflected,

“… you’ve got to just keep yarning, and talking and just keep chipping away … but most of them don’t want to quit smoking. And that’s okay, that’s their choice, but as long as they know the implications and I’m here to help, and let’s just try and reduce, and let’s do little steps at a time …”(case manager)

Second, ESF participants were supported to set their own smoking cessation goals in their own timeframe. As stressors were addressed, participants would re-evaluate their smoking cessation goals, with several participants achieving smoking reductions and/or cessation despite previously considering this to be impossible, as one significant other described,

“Before [ESF], I just saw myself as a smoker … like, when I first heard of [ESF], I didn’t think much of it … I was, I’m not going to really reduce, and then they started to provide support and … I changed my goal, I’ll reduce my smoking … didn’t really think I’d quit … now I’m down to five smokes a day, and my next goal is to quit smoking, so it’s been very successful …”(significant other)

Finally, participants were encouraged to trial different smoking behavioral changes, such as reducing the amount they smoked or their exposure to passive smoking through support to set boundaries around other’s smoking in their personal environments. The case managers and COE health professionals considered this approach both pragmatic and pivotal in enabling participants to begin to imagine a smoke free future. This approach was well received by participants, who described feeling supported without pressure or judgement, 

“… they were willing to work with me and not pressure me about just giving up, you can just slow down in time and I thought that was good … because a lot of people just expect you to quit straight up, not just like want to work with you … we just talked a lot and, just little things that, to avoid smoking and over time I have like cut down heaps …”(pregnant woman)

### 3.4. Individualized Care for ESF Participants in Their Social Context 

With each participant having unique lived experiences and life circumstances, a one-sized fits-all program was inappropriate. Rather, case managers tailored support to each individual, with care plans developed in partnership with the ESF participants. In doing so, participant’s agency and capacity was respected, and their existing strengths and skills were built upon, as one case manager reflected, 

“Letting the participant lead, staying in tune with them and recognizing that they are the lead but I can speak up, act or support them when they need it. Stay in tune with how the different participants work best. Also realize that some participants say they can do it but they might really need more support.”(case manager)

In addressing the social context of women’s smoking, significant others were included in the program, a feature that pregnant women appreciated, as one pregnant woman stated, “because it’s not all on you, it’s on both of you, so the stress is evened out …”. This approach resulted in positive whole-of-family outcomes for some, with one participant expressing,

“Because we were all together for once … [with the program] pushing us to get to our goals … she [significant other] was nothing and now she’s something … I was an alcoholic before I fell pregnant. I was giving up on my daughter, my home, my family …. [and now] I want to do something with myself, fix my teeth, fix myself up, get a car, get a job …”(pregnant woman)

Individualized smoking cessation support that sought to foster supportive and smoke-free environments through providing smoking cessation education and support to women’s families and/or households was also important. As one GP emphasized, 

“… including direct family, including the partners, is essential …. so if you’re just telling one person in a household [trying] to stop smoking, it’s not going to work. But if everyone in that household is talking about the same thing, and getting the same advice, you’ll get the strengths of everyone and you’ll change the conversation in that household or that larger family group.”(GP)

In taking a whole-of-family focus, the program sought to foster a sense of shared responsibility for creating smoke free environments, rather than placing the burden for change solely upon the pregnant woman. The benefit of this was shared by one pregnant woman,

“… because we used to actually smoke in my house … so, yeah, all that changed, because I did live with two smokers, my aunty and my partner, now there’s no smoking in the house … everyone knows not to smoke around me now, because it makes me feel sick too.”(pregnant woman)

Importantly, a tailored and whole-of-family approach was also effective for addressing participants’ other substance use, in particular marijuana. Similar to their tobacco use, marijuana was often used by participants to relieve stress, with the common belief that it was natural and therefore harmless. As one case manager reflected, 

“Their drug use, it’s so hard because it’s embedded … like, the mother smokes yarndi (marijuana), the grandmother, the uncle, the aunty … they see it as their medication … that’s how they kind of survive, so yeah it can be tricky trying to challenge their thoughts and beliefs around yarndi use … but whenever I can I always slip in some little educational message… so over the time when they were engaged his yarndi use went right down and so did hers because they did it together as a team …”(case manager)

### 3.5. Centering Aboriginal and Torres Strait Islander Culture 

In recognizing the foundational place of culture in Aboriginal and Torres Strait Islander peoples’ identity, innate resilience and belonging, culture was centered in ESF through various direct and indirect ways. With the program being integrated within an Aboriginal and Torres Strait Islander primary health care service, Aboriginal and Torres Strait Islander people’s knowledge, vernacular, and ways of relating shaped the space within which care was provided. Two case managers were members of the local Aboriginal and Torres Strait Islander community and known by most ESF participants, with the third non-Indigenous case manager being a trusted long-term worker in the community and sensitive to the local culture and community needs. Indigeneity was also visually celebrated throughout the COE physical space. These factors combined helped foster cultural safety and belonging for ESF participants. 

ESF’s art component also celebrated Aboriginal and Torres Strait Islander culture through a known and trusted local Aboriginal artist doing pregnancy belly painting and photographic portraits. For some pregnant women, pregnancy and the concomitant physical changes were a source of shame rather than joy, sometimes exacerbated by experiences of institutional and interpersonal racism. The program’s art component provided a counterpoint to negative experiences by creating a safe space for participants to celebrate and capture positive memories of pregnancy and family. Participants and case managers highly valued the art component for these reasons, along with the ways in which these activities helped fostered positive connections to participants’ unborn baby and culture, as well as time for relaxation and stress relief - sentiments that the following case manager and participant words reflect,

“… they’re just in survival mode and constantly feeling down about themselves, low self-esteem and stuff like that … […] … you can see [how ESF changed how people saw themselves] in the photos often when you look at them you can tell the Mum’s happy, she’s loving the baby, and it’s a good opportunity for us to have a yarn about what are you going to do about the labor, and talk about names, and really starting to talk about her and the baby.”(case manager)

“An Aboriginal artist doing, working on an Aboriginal person made us feel more connected culturally and confidence in myself in doing that [having their belly painted].”(pregnant woman)

Furthermore, participants described feelings of increased confidence as a result of their participation in the art activities. 

“The art, the belly “[casting] and the photos, I really enjoyed them. I wasn’t a confident person but they, this program has made me come out of my shell. I got my belly [painting] done, I was very proud about it. I gave photos to my families and friends … I’m very private but … I loved doing them and it was good, doing something for myself.”(pregnant woman)

## 4. Discussion

This research has identified that, at its heart, ESF provided relationship-based, holistic, flexible, individualized, and culturally oriented care. These FOV provide important insights into why ESF enabled participants to reduce or cease tobacco use during the pregnancy period, and why the program was positively received by participants and health care providers. In what follows, we review how the FOV fostered the program’s acceptability; created a foundation for smoking reduction or cessation through contextualized support; and in doing so, reoriented the way smoking cessation support was provided with participants. 

### 4.1. Fostering Accessibility 

ESF was relationship-based with a commitment to creating trust and cultural safety for participants. This focus ensured ESF’s accessibility for members of the local Aboriginal and Torres Strait Islander community. Integration of ESF with the COE, a local and trusted Aboriginal and Torres Strait Islander primary health care service, provided a base-level of cultural security from which the case managers developed trusting relationships with participants through dependable, non-judgmental, culturally affirming, and holistic care. A participant-led approach to smoking cessation also supported participants’ personal agency and ownership over their care. Thus, ESF used a trauma-informed approach, where the on-going impact of racism, discrimination, and practitioner bias on Indigenous peoples access to health care is recognized [13,33]. This approach was critical to ESF as the impact of discrimination can be amplified for pregnant Indigenous women who smoke through experiences of stigma and shame around their tobacco use. Consequently, individual behaviorist approaches to smoking cessation become unsafe and psychologically inaccessible—a concerning predicament given the social disadvantage that Indigenous women already face [3,33,34]. In this way, the program’s focus on fostering cultural safety and trusted relationships was not only important for improving the program’s accessibility, but critically, for mitigating the negative impacts of racism and discrimination in healthcare for Aboriginal and Torres Strait Islander people [13]. 

### 4.2. Co-Creating a Foundation for Smoking Reduction and Cessation

Knowing the complex social and structural determinants influencing ESF participants’ tobacco use was vital for addressing their unique constellation of barriers to smoking cessation [3,34,35]. Through a harm-reducing and trauma-informed approach, case managers supported pregnant women and their significant others to reduce complex life stressors through culturally-centered and holistic wraparound care that was tailored and flexible and thereby highly responsive and relevant to participants’ circumstances. Importantly, participants’ innate resilience and strengths, personal agency, and Indigeneity was celebrated. This comprehensive focus marked a critical departure from mainstream practices whereby cessation support often takes the form of standardized advice through brief interventions with narrowed focus on smoking behavioral change without the provision of contextual understanding and support [1,36]. Failing to acknowledge or mitigate the social context and causes of Indigenous women’s smoking has the potential to cause harm by casting the goal of smoking cessation as unattainable [1,37]. This is especially pronounced for Indigenous women, who typically experience a greater intersection of stressors than non-Indigenous women during pregnancy, including experiences of daily stress from social disadvantage, which has been linked to tobacco addiction [38].

Placing Aboriginal and Torres Strait Islander women at the center of the ESF program in this way further contrasts standard cessation support that has narrowly favored fetal health to the exclusion of the contexts impacting women’s health and wellbeing—a focus that risks further stigmatizing women and inducing shame for stress-response behaviors [2,3,39]. The ESF program avoided such harm by being women-centered—focused on progressing the health and wellbeing of pregnant women while recognizing the intrinsic relationship between a woman and her unborn baby’s health [2]. This was made possible through holistic wraparound care that was highly flexible, tailored, and non-stigmatizing, and through including significant others to foster supportive and smoke-free environments to support pregnant women’s smoking behavioral change [2,20,39]. Ultimately, this comprehensive approach to addressing participants’ complex stressors was considered essential for creating a more realistic foundation from which tobacco smoking reduction or cessation could be considered. For many of the pregnant women and their significant others who participated, this foundation took the form of increased housing, food, and income security; increased personal safety from domestic and intimate partner violence; increased access to appropriate mental and physical health care; and increased social and cultural support. Reducing complex life stressors meant that pregnant women had greater capacity to reduce or cease their dependence on not only their use of tobacco for coping with stressors, yet also their use of other substances, including marijuana, alcohol, and in some cases, methamphetamines. 

### 4.3. Journeying Towards Smoking Cessation 

With a strong program focus on addressing the complex stressors that ESF participants faced, participants’ readiness and capacity for smoking reductions or cessation changed over time as stressors were lessened. In response, smoking cessation support needed to be highly attuned and responsive to participants’ changing circumstances. A flexible and tailored approach enabled this to be achieved and was premised on a respect for Aboriginal and Torres Strait Islander people’s authority in determining the ways in which their health and wellbeing can be improved [1]. In delivery, this entailed ESF participants controlling the pace and intensity of their smoking behavioral change efforts and goals, including the ways in which they engaged with NRT. The case managers supported participants’ choices and approached smoking cessation education and support in an unpressured and non-judgmental manner. This process reflected a culturally appropriate approach to motivational interviewing—a process that is recognized for its capacity to achieve higher rates of smoking cessation compared to mainstream approaches delivering standard advice [40]. 

The program’s flexible and tailored approach to smoking cessation support required a reconfiguration of what success looks like in this space—viewing it through a contextual and harm-reduction lens [3,39], rather than a narrow focus where only cessation is valued [1]. Thus, the significance of improvements in ESF participants’ daily circumstances and general health and wellbeing were recognized. Specific to tobacco use, this involved celebrating the success of each change in smoking behavior made, as appropriate to individual circumstances and capacity, including changes in passive smoking exposure, along with reductions or cessation of tobacco use. Supporting ESF participants to be the directors of their smoking cessation journey in a safe environment further exemplified a trauma-informed approach to cessation support by avoiding directive and confrontational approaches to behavioral change goal setting, which can be experienced as unsafe and authoritative, particularly for women who have experienced discrimination and trauma [39].

The ESF approach to smoking cessation demonstrated that when women feel safe and become more supported and secure in their circumstances with their essential needs met, their innate resilience and capacity to improve their health and wellbeing, including making positive smoking behavioral changes, naturally arises. Consequently, the FOV identified here provide a template for progressing all aspects of pregnant Indigenous women’s health and wellbeing during antenatal care, with far reaching benefits for their unborn baby, including reducing women’s use of other substances, such as marijuana, alcohol, methamphetamine, and so on. Indeed, the ESF FOV are strongly echoed in international fetal alcohol spectrum disorder (FASD) prevention frameworks for supporting Indigenous women to reduce or cease alcohol consumption during pregnancy, such as Wolfson et al.’s [41] consensus statement, outlining eight tenets for supporting First Nations, Inuit and Metis peoples in Canada. ESF’ FOV add to a growing international chorus for Indigenous women’s maternal care to be contextualized and holistic, wellness and culture centered, harm-reducing and trauma-informed, and therefore are pertinent for the development of Indigenous women’s healthcare models both beyond tobacco smoking and the Australian contexts.

### 4.4. Strengths and Limitations

Pregnant Indigenous women need to be supported in smoking cessation through intensive and multifactorial interventions that address the social and structural determinants of their health and wellbeing [1,17,20,21]. ESF is one of the few smoking cessation programs that has both applied this approach in culturally affirming ways and explored its merits through both quantitative and qualitative research [18,20,21]. Given the promising nature of the program in supporting positive smoking behavioral changes among its participants, the FOV identified in this study provide critical insights for how smoking cessation programs can support pregnant Indigenous women in safe, pragmatic and effective ways. While this study was situated in one urban Aboriginal and Torres Strait Islander community and therefore generalizations beyond this geographic location are not possible, we contend that the FOV are applicable in other locations and in other programs supporting Indigenous women while pregnant or parenting. Importantly, the FOV are fundamental principles that can be tailored to meet the unique contexts and needs of other communities. It is important to note, however, that while this study has yielded important insights, it was exploratory in nature, and therefore, did not provide systematic evaluation of the ESF intervention, nor did it highlight possible barriers that could be met with the implementation of the FOVs in other locations. Therefore, comprehensive evaluation of any future application of the FOV would further build upon our understanding of the FOVs in practice and would provide valuable contributions to an emerging evidence base for how we can best support Indigenous women who smoke during pregnancy. The importance of this research direction cannot be understated given the persistently high rates of tobacco use among pregnant Indigenous women, despite current efforts [4,8,15,36].

This study used a convenience sampling approach to collect insights into the program’s FOV. With many participants preferring to restrict contact with ESF to when they desired connection with their case manager, it was difficult to reach some participants for the interviewing process. This resulted in contact being limited to those who were most engaged with the program. While this approach to sampling is neither purposeful nor strategic and risks introducing bias into the results [30], there was high degree of concordance in the features of values identified by ESF participants and the health professionals. It is possible, however, that participants who were not interviewed, or who had previously withdrawn from the ESF program may have held differing views. Furthermore, study time constraints limited our ability to gain more nuanced insights into the FOV identified, which could have been achieved through evaluation of the program over a longer time period and conducting subsequent interviews, particularly with pregnant women and their significant others. Regardless of these constraints, rich insights were gained, which we attribute to the pre-established and trusted interviewer–interviewee relationships that enabled participants’ safety to share through the yarning process, which better positioned the interviewer to understand the nuances of participants’ stories.

## 5. Conclusions

ESF’s FOV articulate why it was acceptable and feasible to its participants, case managers and COE healthcare professionals. They contribute nuanced insights into how interventions can foster program accessibility and safety, provide relevant support through supporting women’s health and wellbeing, and critically, provide effective smoking cessation support that is harm-reducing, non-stigmatizing, and trauma-informed. The FOV illuminate a strength-based approach that supports Indigenous women’s agency in culturally safe and affirming ways that are highly responsive to their lived realities, needs and priorities. In doing so, they demonstrate how we can create more humanizing and compassionate support for pregnant Indigenous women and depart from mainstream individualistic behavioralist approaches that risk stigmatizing and shaming those who rely upon smoking for coping with complex stressors. The positive impact of the ESF program has important implications not only for the design and delivery of smoking cessation interventions with pregnant Indigenous women, yet also for the ways in which healthcare professionals, organizations, and smoking cessation messaging can be approached in this space. Beyond smoking cessation, however, the ESF FOV have pertinence for comprehensive approaches to Indigenous women’s antenatal care and the simultaneous addressing of multiple pregnancy health concerns, including risks associated with other substance use.

## Figures and Tables

**Figure 1 ijerph-18-00049-f001:**
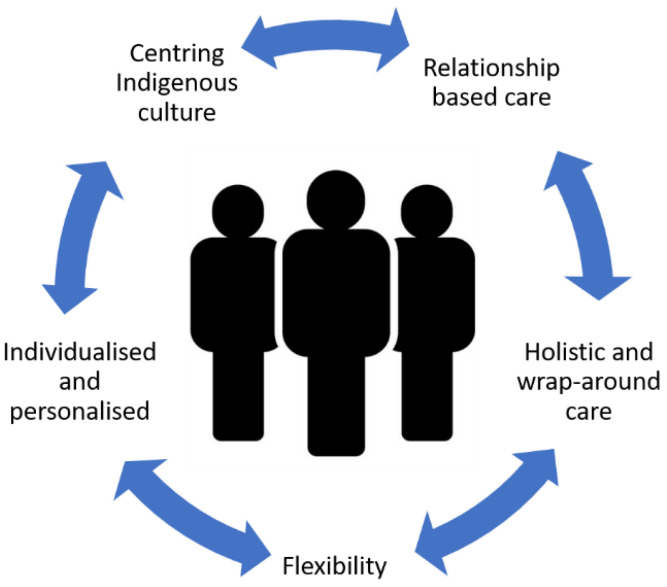
Features of value of the ESF program.

## Data Availability

The data that supports the findings of this study are not publicly available, but are available from the COE, via the corresponding author, provided appropriate ethical and community approvals are obtained.

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
