# Peer review of "“They Were Willing to Work with Me and Not Pressure Me”: A Qualitative Investigation into the Features of Value of a Smoking Cessation in Pregnancy Program for Aboriginal and Torres Strait Islander Women"

_ijerph, 2020, doi:10.3390/ijerph18010049_

Round 1

Reviewer 1 Report

This study presents some informative co-produced data on a contextual and culturally sensitive tailored smoking cessation support programme. The article is very clearly written and compelling.

My only real concern is in the analysis and presentation of results, and how they can be enhanced to inform other contexts and be evaluated systematically. While highlighting features of value is helpful to inform other programmes, it would also be very informative to analyse discrepant findings, especially those which indicate issues/barriers in the programme that could potentially be addressed to improve its delivery.

Moreover, with the sample as it is, it is quite possible that those who did not agree to participate had ideas/reflections that are not captured in this study. The sampling/recruitment for interviews is understandably tricky in hard-to-reach groups, but was there any effort made to explore attributes of the programme that could be improved? It is difficult to gather the extent to which this was possible or relevant.

Some specific comments for each section are below.

Abstract

Although interviewed participants outlined some of the valuable aspects of this programme of support, line 28 arguably overstates the findings on smoking cessation. The authors may consider some more caution when extrapolating the findings from this study and their previous mixed methods study on smoking cessation. From what I understand, 4/11 pregnant women were abstinent at the primary follow-up time point, which is very encouraging but without an appropriate experimental design in this context it is difficult to conclude that the programme was solely responsible for the observed quit success, nor whether this would extend into long-term smoking cessation.

Introduction

Building on previous exploratory work conducted by the authors, a very good introduction covering persistent and sharp inequalities in smoking among aboriginal and Torres strait islander women.

The authors should consider reducing the number of abbreviations in the manuscript. While they may save on the word count, it made some sentences less clear.

Lines 93-98: The study states that it seeks to understand the core features of value of the programme, but then mentions that the aims are to identify the underlying principles of the programme as experienced by participants and practitioners. If the underlying principles of the programme are being explored as experienced, then is not also true of the features of value? i.e. These are features that were identified, but there may be others that were not discussed, or some that would not have been identified as valuable features by other participants?

Methods

Some more information on the yarning method would be helpful. An example of how the interview is introduced and flows using this approach might be considered. Are there any points at which the interviewer needs to guide the conversation?

As mentioned above in my overall comment, why focus only on positive aspects of programme when there may be important limiting factors/barriers that could be addressed to improve the programme?

Without detracting from what are compelling findings, I have a concern with the combination of interviews from participants and practitioners. Is it possible that the practitioners had some bias in how they reflected upon the delivery of the intervention? Linked to this is it possible that those who were willing to be interviewed were more likely to speak favourably of it and highlight aspects that were relevant to them?

The question of data saturation is interesting – what areas were lacking in data or appeared thin from interviews thus far? It would be helpful to have a better idea of the data, perhaps in a supplementary table beyond what is described in the main text of the manuscript. Without this it is hard to know where data were thin.

The integration with other forms of social care is very encouraging, if resource intensive. The flexibility of the programme is clearly a strength, but I am concerned about how feasible such an approach is if implemented at a larger scale. To that end, I wonder whether the researchers were cognizant in thinking about ways to understand key barriers and facilitators to programme implementation, rather than just highlighting areas of value. By informing how the existing programme could be delivered in future, without losing the value of the personalised and holistic approach would be highly informative and support this beyond the piloting stage.

Considerable the overlap between features. Have the authors considered marrying some of the concepts highlighted into a theoretical model such as COM-B/behaviour change wheel? This wouldn’t need to disrupt the bottom-up narrative approach listed here or change the overall findings, but rather highlighting the points of difference within each themes with respect to automatic and reflective motivation, physical and social opportunity, and psychological and physical capability. By mapping some of the themes onto a framework you may be able to inform future intervention development with a theoretical underpinning, while also being able to highlight specific barriers and facilitators of the programme. This would also be useful when evaluating the programme against other forms of tailored support, where specific intervention factors can be evaluated. An example is below in a very similar context, which the authors may be familiar with:

Gould, G. S., Bar-Zeev, Y., Bovill, M., Atkins, L., Gruppetta, M., Clarke, M. J., & Bonevski, B. (2017). Designing an implementation intervention with the Behaviour Change Wheel for health provider smoking cessation care for Australian Indigenous pregnant women. Implementation science, 12(1), 114.

Discussion

An excellent point regarding other substance use in disadvantaged communities. Very often support is viewed in vertical terms (i.e. smoking without consideration of other behaviours when they are closely related).

Is the intervention being evaluated in a larger RCT, as mentioned in the authors mixed methods paper? If so some information on the potential for this would be useful.

Minor comments/typos

Final paragraph line 518 typo in sentence. Presume should be “…approached in this space” rather than “…approached this space”?

Reviewer 2 Report

See attached

Reviewer 3 Report

It is an interesting study about the ESF program in aboriginal pregnancy women

and the relationship between smoking cessation and life experiences

Some further details should be provided about people recruited:

did recruited people had any comorbidities?

Please include a sentence about any clinical symptoms reported

In the methods section please include a statistical approach used.

data regarding nicotine dependence should be included as far as  possible.

Were e-cigarette used by pregnancy people? please include a statement about

I suggest to include the following references reporting nicotine dependence test in smokers and smoking initiation in aboriginal people:

Clin Respir J. 2020 Jan;14(1):29-34. doi: 10.1111/crj.13096.

Aust N Z J Public Health. 2020 Oct;44(5):397-403. doi: 10.1111/1753-6405.13022

Round 2

Reviewer 1 Report

The authors have responded to my questions and clarified the main issues I raised. I have no further comments to add.